# Global epidemiology and spatial distribution of *Toxoplasma gondii* in goats: Protocol for a systematic review and Bayesian hierarchical meta-analysis

**Afsaneh Amouei**[1,2], **Azadeh Mizani**[3], **Ahmad Ali Hanafi-Bojd**[4], **Tohid Jafari-Koshki**[5], **Shahabeddin Sarvi**[1,6], **Sargis A. Aghayan**[7], **Fateme Amuei**[8], **Tooran Nayeri Chegeni**[1,6], **Ahmad Daryani**[1,6]*

**1** Toxoplasmosis Research Center, Communicable Diseases Institute, Mazandaran University of Medical Sciences, Sari, Iran, **2** Mazandaran Central Veterinary Laboratory, Medical Sciences, Veterinary Administration of Mazandaran Province, Sari, Iran, **3** Department of Parasitology, Pasteur Institute of Iran, Tehran, Iran, **4** Department of Medical Entomology &Vector Control, School of Public Health, Tehran University of Medical Sciences, Tehran, Iran, **5** Department of Statistics and Epidemiology, Faculty of Health, Tabriz University of Medical Sciences, Tabriz, Iran, **6** Department of Parasitology, School of Medicine, Mazandaran University of Medical Sciences, Sari, Iran, **7** Laboratory of Zoology, Research Institute of Biology, Yerevan State University, Yerevan, Armenia, **8** Department of Organic Chemistry, University of Mazandaran, Babolsar, Iran

* daryanii@yahoo.com

**🔓 OPEN ACCESS**

## Abstract

### Background

*Toxoplasma gondii*, a cosmopolitan protozoan parasite causes toxoplasmosis in humans and many species of domestic and wild animals. *T. gondii* instigates significant economic losses in sheep and goat farming industry and can lead to abortion, stillbirth, congenital malformations and neonatal losses. The objective of this protocol is to evaluate worldwide seroprevalence of *T. gondii* exposure in goats using Bayesian hierarchical meta-analysis and geographic information system (GIS).

### Methods

A comprehensive literature search will be conducted using search engines, including Web of Science, ScienceDirect, Scopus, PubMed, ProQuest, EMBASE, PROSPERO Register and, Google Scholar without date and language restrictions. The authors search for cross-sectional studies that determine the seroprevalence of *T. gondii* in goats. Two reviewers will independently screen, selected studies; also, they will extract data, and assess the risk of bias. In case(s) of disagreement, a consensus will be reached with the help of a third author. The Bayesian hierarchical meta-analysis will use to estimate country and worldwide true seroprevalence of *T. gondii*, which is consist of the sensitivity and specificity of the applied serological assays. The obtained data will be used to identify country-level risk factors associated with *T. gondii* exposure using GIS in the ArcGIS software.

**Data Availability Statement:** Deidentified research data will be made publicly available when the study is completed and published.

**Funding:** The authors received no specific funding for this work.

**Competing interests:** The authors have declared that no competing interests exist.

## Discussion

The systematic review produced from this protocol will provide the true prevalence rate and spatial distribution *T. gondii* exposure in goats both regionally and globally using Bayesian hierarchical and GIS analysis.

## Systematic review registration

PROSPERO registration number: CRD42020107928.

## Introduction

*T. gondii* is one of the most well studied coccidian parasites and causes widespread infections in almost all warm-blooded animals including humans and livestock. Parasite also plays a considerable zoonotic role and has both veterinary and medical importance worldwide [1, 2]. The infection is usually asymptomatic in humans but can also cause severe complications in immunocompromised individuals and abortion. Parasite may lead to abortion in ruminants, cause huge economic losses to the livestock industry and facilitate the transmission of infectious and parasitic disease to humans, due to the consumption of infected raw or undercooked meat or milk [3–5].

Goats are one of the main sources of meat borne infection [6] and play an essential role in the economy of some countries [7] since they are significant sources of their products (meat and milk). Toxoplasmosis is globally recognized as a food-borne disease and a public health problem. Researchers reported the occurrence of toxoplasmosis in humans and goats, which was linked to the consumption of raw goat milk and milk products [8, 9]. Based on data obtained in some regions, the seroprevalence of *T. gondii* can be as high as 77% in goats [4]. It is estimated that 30% to 63% of human infections have been exposed to *T. gondii* [8]. In Europe, among 14 foodborne diseases, toxoplasmosis has the highest human incidence and in the USA was suggested as one of three pathogens (together with *Salmonella* and *Listeria*) [10, 11]. However, human infections from different regions of the world depend on prevalence the of parasite in animals and eating habits [12].

Numerous reports from epidemiological surveys have been conducted on animal toxoplasmosis, although globally, no exhaustive documented data are available for the prevalence of goat toxoplasmosis. The aims of the current meta-analysis, therefore, were, (i) to estimate the true prevalence of *T. gondii* among goats in the world in a Bayesian framework, (ii) to assess its association with several risk factors and to display the distribution of *T. gondii* exposure in different countries using GIS.

## Materials and methods

### Study design

This systematic review will be conducted according to the Preferred Reporting Items for Systematic Reviews and Meta-Analyses Protocol (PRISMA-P) guidelines [13] when reporting the findings (S1 Checklist). The current study protocol was registered with PROSPERO (https://www.crd.york.ac.uk/prospero/display_record.php?ID=CRD42020107928) [14]. The systematic review with meta-analysis will be managed following the recommendations of the Cochrane Collaboration Handbook of Systematic Reviews [15], as a systematic, transparent and reproducible method to investigate of scientific evidence.

**Table 1. Eligibility criteria.**

| Category | Inclusion | Exclusion |
|---|---|---|
| Population | Studies that report prevalence of *Toxoplasma gondii* exposure in goats based on serologic assay | Studies that report prevalence of *Toxoplasma gondii* exposure in goats based on molecular methods |
| Exposure | Goat toxoplasmosis | |
| Comparator | None | |
| Outcomes | True seroprevalence of goat *Toxoplasma* exposure country and worldwide and its spatial distribution | |
| Study Type | Cross-Sectional | |

## Eligibility criteria

The eligibility criteria for the study were determined using the PICOS classification (Population, Intervention, Comparison, Outcome, and Study design) as a tool to guide the research and adjust the search strategy, as described in Table 1 [16]. No language and date restrictions were applied in this study.

## Search strategy

To assess the seroprevalence of *T. gondii* goats from a global perspective, six databases, namely the PubMed, Scopus, ScienceDirect, Web of Science, EMBASE, ProQuest, PROSPERO Register, and the Internet search engine Google Scholar, will be searched by two authors (AA and AM) of a trained team for relevant studies published before June 2022. All databases will be searched using the keywords terms presented in Table 2 in all fields. Keywords will use individually or combined with a list of all of them and without any language restrictions. The authors systematically scrutinized a manual search of the reference lists of the identified reports and reviews.

## Study selection

All cross-sectional studies will be included if they meet the inclusion criteria. Two reviewers of the research team will independently review the retrieved papers. All search records will be imported to EndNote software V.X7.0.1 software. Following the removal of duplicate entries, each of these studies will be screened, by reading the title and the contents of the abstract or the full-text to identify papers that described studies. A data extraction sheet will then be drawn and data will be recorded in the selection sheet. Discrepancies will be resolved with the provision for arbitration from a third reviewer within the review team. Study selection will be performed according to the PRISMA flowchart (Fig 1) [17].

**Table 2. Search strategy.**

| Search | Search terms and combinations |
|---|---|
| 1 | *Toxoplasma gondii* OR Toxoplasmosis OR Toxoplasmosis in animal |
| 2 | Meat producing animal OR Goat OR Caprine OR Livestock OR Ruminant |
| 3 | Prevalence OR Seroprevalence OR Seroepidemiology OR Epidemiology |
| 4 | Cross-sectional study |
| 5 | Evaluation OR Detection |
| 6 | #1 AND #2 AND #3 AND #4 AND #5 |

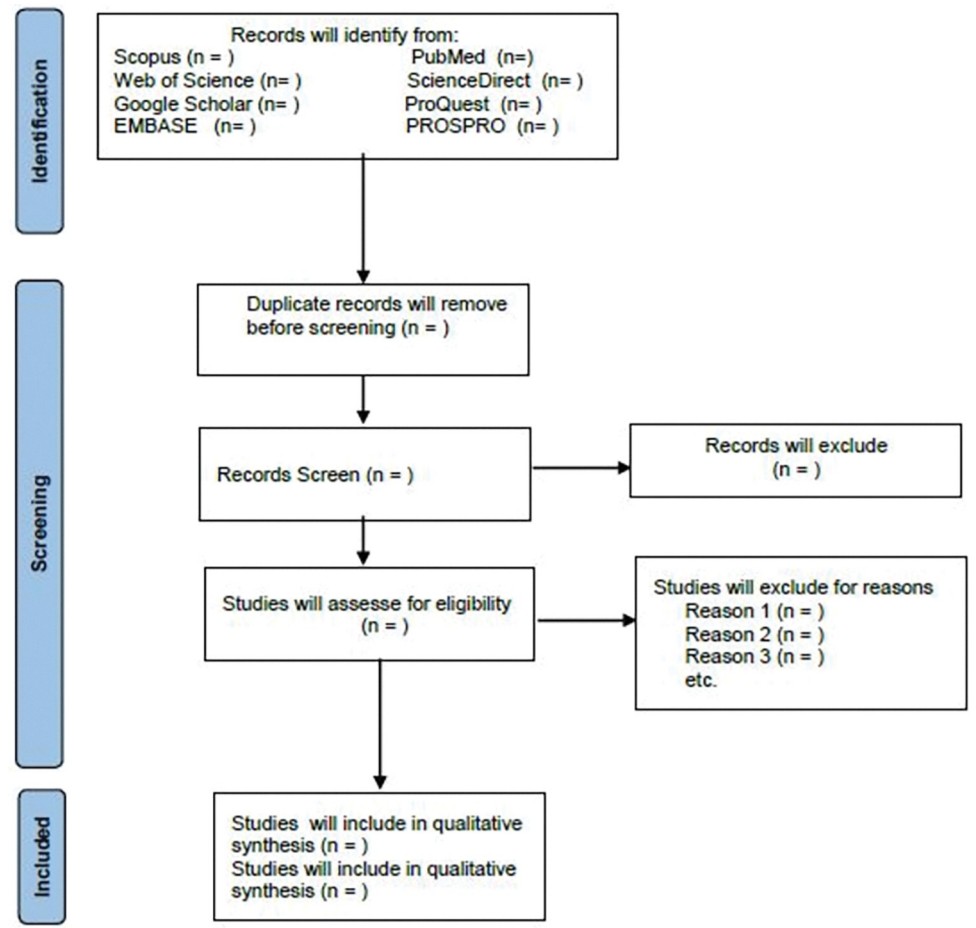

**Fig 1. PRISMA protocol flowchart of literature selection.**

## Data extraction

Data from the retrieved studies will be collectd independently by two reviewers, using a piloted form which will be checked by another investigator. Abstracts will not be included in the meta-analysis for risk factors. Finally, data will be transferred to a Microsoft Excel software (version 2016; Microsoft, Redmond, WA, USA). The following information will be used (S2 Checklist): title, first author, year of publication, continent, country, total sample size, number of positive samples, gender, age group ($< 1$ year and $\geq 1$ year), detection method, the presence or absence of cats on the farm, system of management (extensive, intensive, and semi-intensive), sensitivity, specificity, as well as geographic and natural climatic conditions for each country (latitude, longitude, altitude, mean annual temperature, relative humidity, and annual rainfall). Data will also be collected for both minimally adjusted and maximally adjusted risk estimates, if available (presence of cat, farm management age, and gender adjustment). The inclusion criteria are: 1) studies published until June 10[th], 2021, 2) cross-sectional studies that estimate the prevalence of toxoplasmosis in goats, 3) original research studies, 4) papers with available full texts, and 5) studies with information on the total sample size and positive samples. The exclusion criteria are: 1) studies with no exact information about the sample size and the diagnostic criteria, 2) descriptive, letters to the editors, and review articles, 3) studies conducted in other animal and human models.

## Assessment of bias

A formal assessment for the risk of bias in the included reports will have limited utility given the lack of an appropriate assessment tool in animal prevalence studies. Although quality checklists have been developed for meta-analyses studies of disease prevalence, many aspects of the available checklist are not straightly related to the present survey question [18]. Therefore, study quality will be assessed based on 10 key factors formulated as questions which will score 3, 2, or 1 based on a simple scale system (3 for "yes", 2 for "no", or 1 for "unsure") [19]. These questions are as follows:

Q1: Is the purpose of the study defined well?
Q2: Is the target population (time and location) defined well?
Q3: Are the inclusion and exclusion criteria defined well?
Q4: Is the sampling method (random sampling) specified well?
Q5: Is the sample size adequate?
Q6: Is the working method described well?
Q7: Is an appropriate diagnostic method used?
Q8: Are the subgroups divided well?
Q9: Is a proper analysis does well?
Q10: Are the effects of confounders removed well?

A quality score will determine by rescaling the sum of scores of each eligible paper between 10 to 30 points with a low-quality report earning a score of $\geq 15$. Quality assessment will be completed independently by two assessors, and a table of quality score computation for each eligible paper will be addressed in the S2 Checklist.

## Data analysis

The data entered into Microsoft Excel will be analyzed using the STATA statistical software (version 14; Stata Corp, College Station, TX, USA). The collected data will be analyzed using the Bayesian hierarchical meta-analysis model as follows. First, the number of sero-positive cases in the i-th country, $x_i$, grouped according to the continents, will be assumed to follow a binomial distribution with a country-specific apparent seroprevalence $AP_i$. Logit-transformed seroprevalence in each country will be assumed to follow a normal distribution with a common mean over the pertinent continent, that is, logit $(AP_i) \sim$ Normal $(\theta_j, \sigma_w^2)$, where i represents the country and j represents its continent and $\sigma_w^2$ is the variance of seroprevalence over the continent. We will assume normally distribute hyperpriors for the continent mean seroprevalence, $\theta_j \sim$ Normal $(\theta_0, \sigma_b^2)$ in which $\theta_0$ is the worldwide mean and $\sigma_b^2$ is the variance of seroprevalence of *T. gondii* among continents. After the estimation of the Bayesian posteriors, the logit-transformed missing seroprevalence of *T. gondii* for study k in a continent, logit $(\theta_k^*)$, will be estimated using the posterior predictive distributions. That is, the missing data will be generated from a normal distribution as $\text{logit}(\theta_k^*) \sim \text{Normal}(\hat{\theta}_0, \hat{\sigma}_b^2)$.

The next step will include the estimation of the true seroprevalence of each study ($TP_i$). To conduct this analysis, data on sensitivity (Se) and specificity (Sp) of commercial kits used in these studies will be collected. This information will be considered in their contribution to estimating the true seroprevalence, $TP_i$, by the equation below.

$$AP_i = Se_i \times TP_i + (1 - SP_i) \times (1 - TP_i)$$

In the Bayesian setting, due to their possible data range, Beta-distributed priors will be used for Se and Sp [20]. As before, a normal distribution will be considered for logit-transformed

true seroprevalence; $\text{logit}(TP_i) \sim \text{Normal}(\pi_j, \sigma_w^2)$, with normal hyperpriors $\pi_j \sim \text{Normal}(\pi_0, \sigma_a^2)$.

In all models, a Cauchy distribution will be assumed for variances. All models will be run in the R-interface of Stan Bayesian modeling language [21]. The convergence of models will be assessed by inspecting history, autocorrelation plots, R-hat and the number of effective sample sizes. Posterior estimates of parameters will be reported with 95% credible intervals.

## Spatial epidemiology

Data from this the systematic review will be used to identify country-level risk factors associated with *T. gondii* exposure using GIS. In the ArcGIS software, the spatial distribution of this parasite will be mapped. The MaxEnt model (MaxEnt software version 3.4.1 [22] will be used to predict the distribution of *T. gondii*. The following six climatic factors and four environmental variables will be used:

Altitude as the Elevation above the sea level (m), BIO1 as the Annual Mean Temperature (˚C), BIO2 as the Mean Diurnal Range [Mean of monthly (max temp—min temp) (˚C)], BIO3 as the Isothermality (BIO2/BIO7) (×100), BIO7 as the Temperature Annual Range (BIO5-BIO6) (˚C), BIO12 as the Annual Precipitation (mm).

The climatic variables and altitude will be downloaded from the two WorldClim website at a resolution of 1 km$^2$. A set of global climate layers with a spatial resolution of about 1 km$^2$ will be download from the online free WorldClim platform (https://www.worldclim.org). World-Clim (version 2) will be used for average monthly climate-data (minimum, mean, and maximum temperature and precipitation). All points' coordinates will be manually put into the nasa.gov system (https://power.larc.nasa.gov/data-access-viewer/) and inter-annual values will be extracted for precipitation, humidity and temperature.

In total 80% of collection points will be used randomly by MaxEnt for model training and 20% will be kept for testing the results. Finally, the output of the model will identify the environmental suitability for the presence of parasite. Jackknife analysis of the MaxEnt model will be used to find which climatic and/or environmental factors have the greatest effect on the distribution of *Toxoplasma*.

## Ethics and dissemination

Ethical approval is not required since this protocol is for systematic review and meta-analysis. Final reports of this study will be disseminated in peer-reviewed journals and will be made available through conference proceedings.

## The status and timeline of the study

This systematic review and meta-analysis are ongoing. We expect to complete it and will be reported within 12 months.

## Discussion

*T. gondii* is a protozoan parasite that infects a wide range of vertebrate hosts, such as birds, wild and domestic animals and humans. Among domestic animals, goats are the most susceptible host to *T. gondii*. Infection caused this parasite can lead to pre-term deliveries, abortion, weak newborns, and death in young or adult animals and decreased milk production, diarrhea and hair opacity among symptomatic animals [23].

The proposed protocol has several strengths. Firstly, it will adhere to the guidelines, the studies published in multiple languages (if available), and several researchers will

independently perform the intrinsic evaluation method. Secondly, using Bayesian hierarchical and GIS analysis, the true prevalence of *T. gondii* exposure in goats will be systematically reviewed regionally and globally. The Bayesian hierarchical model used in this study will allow to incorporate the sensitivity and specificity of the serological test.

Thirdly, we will undertake the geographical distribution of exposure in a cross-sectional serological study, including prediction modeling using GIS. The present study has some limitations as well: (i) the study design and between-study heterogeneity; (ii) the restriction of research to unpublished, ongoing, gray literature; (iii) the use of diverse diagnostic methods with variations in the sensitivity and specificity and iiii) the use of reports with limited information on some of risk factors. We also believe that these limitations will be justified by meta-regression analysis on the adjusted seroprevalence and a regression model implemented within a Bayesian hierarchical framework. Nevertheless, from a global perspective, our multidimensional approach report here will be very close to true parasite seroprevalence in goats.

The authors hope that the spatial distribution of *Toxoplasma* prevalence in meat-producing animal will be helpful in developing effective intervention strategies to reduce the burden of this zoonotic disease. In addition, we expect that using spatial epidemiology and GIS technology will map diseases to identify the known distribution of pathogens.

## Supporting information

**S1 Checklist. PRISMA-P 2015 checklist.**
(DOCX)

**S2 Checklist. Data extraction form.**
(XLSX)

## Acknowledgments

Authors express their gratitude on the Research Deputy and Technology and Toxoplasmosis Research Center (TRC), Mazandaran University of Medical Sciences, Sari, Iran, to approve the research project No. 14522.

## Author Contributions

**Conceptualization:** Afsaneh Amouei, Ahmad Daryani.

**Data curation:** Afsaneh Amouei, Azadeh Mizani, Fateme Amuei, Tooran Nayeri Chegeni.

**Formal analysis:** Ahmad Ali Hanafi-Bojd, Tohid Jafari-Koshki, Sargis A. Aghayan.

**Investigation:** Afsaneh Amouei, Azadeh Mizani, Shahabeddin Sarvi.

**Methodology:** Afsaneh Amouei, Azadeh Mizani, Ahmad Ali Hanafi-Bojd, Tohid Jafari-Koshki, Shahabeddin Sarvi, Sargis A. Aghayan, Ahmad Daryani.

**Project administration:** Ahmad Daryani.

**Software:** Ahmad Ali Hanafi-Bojd, Tohid Jafari-Koshki, Sargis A. Aghayan, Fateme Amuei, Tooran Nayeri Chegeni.

**Supervision:** Ahmad Daryani.

**Validation:** Afsaneh Amouei, Azadeh Mizani, Ahmad Daryani.

**Writing – original draft:** Afsaneh Amouei, Azadeh Mizani.

**Writing – review & editing:** Ahmad Ali Hanafi-Bojd, Tohid Jafari-Koshki, Ahmad Daryani.

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
