## [Decision Letter · Decision Letter 0]

1 Aug 2023

Global epidemiology and spatial distribution of Toxoplasma gondii in goats: Protocol for a systematic review and Bayesian hierarchical meta-analysis

PONE-D-23-21260

Dear Prof. Daryani,

We’re pleased to inform you that your manuscript has been judged scientifically suitable for publication and will be formally accepted for publication once it meets all outstanding technical requirements.

Kind regards,

Bibi Razieh Hosseini Farash

Academic Editor

PLOS ONE

   "The funders did not and will not have a role in study design, data collection and analysis, decision to publish, or preparation of the manuscript" 

e) Please provide an amended Funding Statement that declares *all* the funding or sources of support received during this specific study (whether external or internal to your organization) as detailed online in our guide for authors at http://journals.plos.org/plosone/s/submit-now.  

f) Please state what role the funders took in the study.  If any authors received a salary from any of your funders, please state which authors and which funder. If the funders had no role, please state: "The funders had no role in study design, data collection and analysis, decision to publish, or preparation of the manuscript." 

Please send your amended statements by return email; we will change the online submission form on your behalf. 

Reviewers' comments:

Reviewer's Responses to Questions

**Comments to the Author**

1. Does the manuscript provide a valid rationale for the proposed study, with clearly identified and justified research questions?

Reviewer #1: Yes

Reviewer #2: No

2. Is the protocol technically sound and planned in a manner that will lead to a meaningful outcome and allow testing the stated hypotheses?

Reviewer #1: Yes

Reviewer #2: Yes

3. Is the methodology feasible and described in sufficient detail to allow the work to be replicable?

Reviewer #1: Yes

Reviewer #2: Yes

4. Have the authors described where all data underlying the findings will be made available when the study is complete?

Reviewer #1: Yes

Reviewer #2: Yes

5. Is the manuscript presented in an intelligible fashion and written in standard English?

Reviewer #1: Yes

Reviewer #2: Yes

6. Review Comments to the Author

You may also provide optional suggestions and comments to authors that they might find helpful in planning their study.

Reviewer #1: I read the manuscript and confirm the structures. there is no optional suggestions and comments to authors that they might find helpful in planning in this study.

Reviewer #2: - This protocol meta-analysis can be a guide for other author to write meta-analysis with data.

7. PLOS authors have the option to publish the peer review history of their article (what does this mean?). If published, this will include your full peer review and any attached files.

Reviewer #1: No

Reviewer #2: No

---

## [Editor Report · Acceptance letter]

16 Aug 2023

PONE-D-23-21260 

Global epidemiology and spatial distribution of *Toxoplasma gondii* in goats: Protocol for a systematic review and Bayesian hierarchical meta-analysis 

Dear Dr. Daryani:

I'm pleased to inform you that your manuscript has been deemed suitable for publication in PLOS ONE. Congratulations! Your manuscript is now with our production department. 

Kind regards, 

on behalf of

Dr. Bibi Razieh Hosseini Farash 

Academic Editor

PLOS ONE